# The aesthetic value of reef fishes is globally mismatched to their conservation priorities

**Juliette Langlois**[1☉], **François Guilhaumon**[1,2], **Florian Baletaud**[1], **Nicolas Casajus**[3], **Cédric De Almeida Braga**[4], **Valentine Fleuré**[1], **Michel Kulbicki**[5], **Nicolas Loiseau**[1], **David Mouillot**[1,6], **Julien P. Renoult**[7], **Aliénor Stahl**[8], **Rick D. Stuart Smith**[9], **Anne-Sophie Tribot**[10,11], **Nicolas Mouquet**[1,3☉]*

1 MARBEC, Univ Montpellier, CNRS, Ifremer, IRD, Montpellier, France, 2 UMR 9220 ENTROPIE, IRD, Université de la Réunion, Université de la Nouvelle-Calédonie, IFREMER, CNRS, La Réunion, France, 3 FRB–CESAB, Montpellier, France, 4 Orixas, Rennes, France, 5 UMR Entropie—IRD—Université de Perpignan, Perpignan, France, 6 Institut Universitaire de France, 1 rue Descartes, Paris, France, 7 CEFE, UMR 5175, CNRS, Univ Montpellier, University Paul Valery Montpellier, EPHE, Montpellier, France, 8 Department of Biology, Concordia University, Montreal, Quebec, Canada, 9 Institute for Marine and Antarctic Studies, University of Tasmania, Hobart, Tasmania, Australia, 10 MIO, Univ Aix-Marseille, Univ Toulon, CNRS, IRD, Marseille, France, 11 UMR TELEMMe, Univ Aix-Marseille, CNRS, Aix-en-Provence, France

☉ These authors contributed equally to this work.
* nicolas.mouquet@cnrs.fr

**Data Availability Statement:** Most of the data used in this paper are freely available and downloadable from the web. Data on IUCN threat status are available in the IUCN Red List database

## Abstract

Reef fishes are closely connected to many human populations, yet their contributions to society are mostly considered through their economic and ecological values. Cultural and intrinsic values of reef fishes to the public can be critical drivers of conservation investment and success, but remain challenging to quantify. Aesthetic value represents one of the most immediate and direct means by which human societies engage with biodiversity, and can be evaluated from species to ecosystems. Here, we provide the aesthetic value of 2,417 ray-finned reef fish species by combining intensive evaluation of photographs of fishes by humans with predicted values from machine learning. We identified important biases in species' aesthetic value relating to evolutionary history, ecological traits, and International Union for Conservation of Nature (IUCN) threat status. The most beautiful fishes are tightly packed into small parts of both the phylogenetic tree and the ecological trait space. In contrast, the less attractive fishes are the most ecologically and evolutionary distinct species and those recognized as threatened. Our study highlights likely important mismatches between potential public support for conservation and the species most in need of this support. It also provides a pathway for scaling-up our understanding of what are both an important nonmaterial facet of biodiversity and a key component of nature's contribution to people, which could help better anticipate consequences of species loss and assist in developing appropriate communication strategies.

(https://www.iucnredlist.org/). RLS data for species lists and some trait information are available through an online portal accessible through (http://www.reeflifesurvey.com), with additional trait data available on request by using the contact form. For each species, we provide aesthetic values predicted in the present study (S3 Data) and web links to original photographic material (S4 Data). Images free of copyright can be provided on request. Other datasets used in this study (extraction from the online survey and images features analysis) and all code used for the analysis and figures are available from the GitHub Repository: https://github.com/nmouquet/RLS_AESTHE.

**Funding:** This research was partially funded through the 2017–2018 Belmont Forum and BiodivERsA REEF-FUTURES project under the BiodivScen ERA-Net COFUND program with the French National Research Agency (DM and NM). This project received additional funding from the LabEx CeMEB and the program PEPS CNRS (NM). RLS data management is supported by Australia's Integrated Marine Observing System enabled by the National Collaborative Research Infrastructure Strategy (RSS). The funders had no role in study design, data collection and analysis, decision to publish, or preparation of the manuscript.

**Competing interests:** The authors have declared that no competing interests exist.

**Abbreviations:** CNN, convolutional neural network; ED, Evolutionary Distinctiveness; GBIF, Global Biodiversity Information Facility; GLMM, generalized linear mixed model; IUCN, International Union for Conservation of Nature; LC, Least Concern; NE, Not Evaluated; NCP, nature's contribution to people; PCA, principal component analysis; RLS, Reef Life Survey; TH, Threatened.

## Introduction

Numerous nonmaterial facets of biodiversity comprise important components of nature's contribution to people (NCP) [1,2]. Among these, aesthetic value (or less formally called "beauty") is one of the most direct emotional links humans can experience with nature [3,4] and can occur through direct (first-hand experience) but also indirect mechanisms (for example, social media, television). It therefore engages a broader cross-section of the human population than most other NCP, but remains relatively poorly studied. The implications of aesthetic value for biodiversity conservation are likely to be substantial [5]. This lack of scientific attention is probably, at least in part, associated with a difficulty in defining aesthetic value [6], and an associated difficulty in consistently and quantitatively measuring the aesthetic value of biodiversity [5,7]. While also applicable to communities and ecosystems, the aesthetic value of individual species is the simplest and most intuitive unit of measurement for understanding this form of connection between humans and nature. Measuring species' aesthetic value thus remains an important step in better understanding and predicting the willingness and motivation of societies to protect species, and the reasons behind success or failure of conservation efforts [8,9].

Biases in research and conservation efforts have been documented for many taxa. For example, vertebrates are far better represented than invertebrates among articles published in conservation journals [10,11] and in biodiversity datasets [12]. More than half of the billions of occurrences reported in Global Biodiversity Information Facility (GBIF) are for birds, while they represent only 1% of the total number of species in GBIF [12]. These biases are explained by human preferences for particular taxa [13–15], with aesthetic value being an important underlying factor in these preferences. For example, fishes and birds displaying bright colors are considered more beautiful by the general public [16,17] and are more likely to be identified and reported in databases of public observations. Such bias is not limited to data collected by the public; Bellwood and colleagues [18] found evidence for potential subconscious bias towards yellow fishes in the published literature on coral reefs.

Previous studies working on aesthetic value have used either expert knowledge or public surveys based on photographic datasets [19]. Such studies are resource and time intensive, and thus they have been limited to a small number of species. Alternative approaches tried to use visual features (that is, pattern analysis, color distribution) in images known to positively influence aesthetic value [20]. These methods are promising but assume prior knowledge on which features, among a myriad, contribute to the aesthetic value and on their relative importance when it comes to compute a single index. Machine learning models, specifically convolutional neural networks (CNNs), have recently become advanced enough to accurately identify patterns (classification tasks) and predict continuous variables on images [21], opening up a new avenue for investigating the visual perception of our environment, without needing to assume or define important features a priori. CNN have already successfully assessed the beauty of outdoor places [22] and coralligenous reefs [23] but have not yet been applied to species level.

Here, we used traditional photographic surveys augmented by a CNN approach to evaluate the aesthetic value of the world's reef ray-finned fish fauna. Our photographic survey, including 13,000 respondents from the broad public, generated a learning dataset for the CNN that allowed estimation of the aesthetic value for 2,417 reef fish species with high predictive accuracy, based on 4,881 species images, and without a priori knowledge on the features which contribute to fish beauty.

We chose to evaluate reef fishes because this group is emblematic of highly valued reef ecosystems and is known for its exceptional morphological diversity [18], which is presumably associated with a large range of aesthetic values. Reef fishes are also essential to the functioning

of some of the most important and endangered ecosystems on Earth [24,25], and are of vital importance for a large part of humanity, including the poorest [26,27], by supporting several economic activities like subsistence fishing, recreational scuba diving, and aquarium trade [24,26,28]. We used data from the widespread and standardized Reef Life Survey (RLS) program [29] to objectively select from all of the world's reef fish species to a more manageable collection of those most commonly encountered by divers, providing us with a subset of 2,417 species. We then mapped fish aesthetic value across the Tree of Life, and considered ecological traits, IUCN threat categories, and importance for fisheries to better understand the potential implications of human aesthetic bias for reef fish conservation.

## Results

### Building the CNN training dataset

Our first task was to build a set of fish images for direct evaluation of aesthetic value by humans that could be used for training a CNN (Fig 1A). We combined a set of 157 fish images previously evaluated [16] with a new set of 345 images independently evaluated in an online survey (see Methods and Text A and B in S1 File). The online survey presented images in pairs

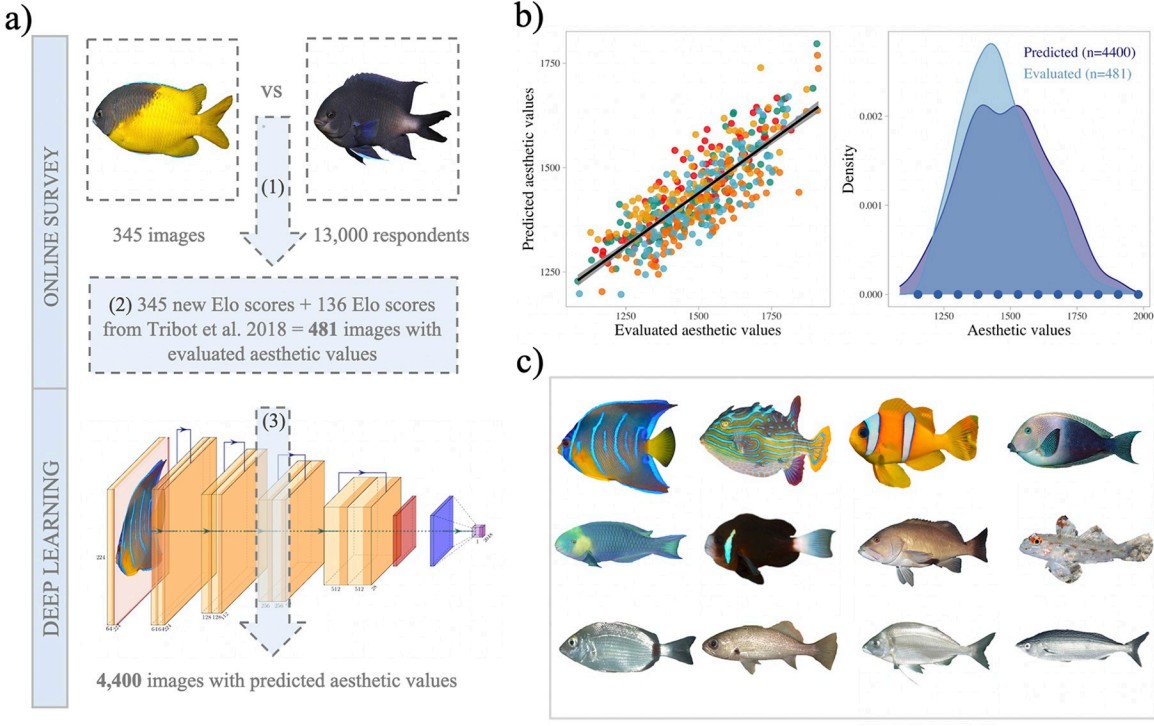

**Fig 1. Evaluation and prediction of fish aesthetic values.** (a) Workflow of the online survey and deep learning prediction of aesthetic values. (1) Pairs of images were presented to the public during the online survey and scored using the Elo algorithm (see Methods). Left *Parma bicolor* and right *Abudefduf luridus*. (2) Once the 345 new images were evaluated online, the values of the 157 images previously evaluated [16] were corrected using the 21 images shared between the 2 surveys. (3) The resulting 481 images with evaluated aesthetic values were used to train a ResNet50 algorithm (see Text E and Fig L in S1 File). Illustration inspired from the PlotNeuralNet [31]. (b) Left: The $r^2$ of the linear relationship between the predicted values averaged across the 5 validation sets and the evaluated values is 0.79 ± SD 0.04 (the color of points indicates the 5 sets used to perform the cross validation). This algorithm was used to predict the aesthetic values of the 4,400 unevaluated images of our dataset. Right: Distribution of the 481 evaluated values in light blue and of the 4,400 predicted aesthetic values in dark blue. The dots at the bottom of the plot indicate the predicted aesthetic values of the images shown in panel (c). Data and code required to generate this Figure can be found in https://github.com/nmouquet/RLS_AESTHE. (c) Examples of fishes representative of the range of predicted aesthetic values. Decreasing aesthetic value from left to right and top to bottom: *Holacanthus ciliaris*, *Aracana aurita*, *Amphiprion ephippium*, *Ctenochaetus marginatus*, *Scarus spinus*, *Amphiprion bicinctus*, *Epinephelides armatus*, *Fusigobius signipinnis*, *Diplodus annularis*, *Odontoscion dentex*, *Nemadactylus bergi*, *Mendosoma lineatum*. See S1 Data for image copyright.

to the public (hereafter called "respondents") and asked to choose the image they found the most beautiful. For the analyses, we kept 13,000 respondents without self-reported color vision issues (see Text C in S1 File). We then estimated each fish aesthetic value through Elo scores computation, a rating system based on pairwise comparison (see Methods) [30].

We tested the potential effect of respondents' sociocultural background and geographic origins on their selections (see Text C in S1 File) using a backward sequential selection procedure with a generalized linear model (see Methods). We found no significant effect of any of the considered variables on selections; we thus computed the Elo scores by pooling across the 13,000 respondents. Among the 345 images evaluated in the new survey, 21 were deliberate duplications from a previous survey [16], included to test for consistency with previous results and increase our learning dataset. For these 21 images, we found a strong correlation between the 2 evaluations ($r^2$ = 0.89, $p$-value < 0.001, Text D in S1 File) and used this relationship to correct the scores of the 157 images previously evaluated [16] and pool the 2 datasets. This resulted in a combined dataset of 481 images with empirically evaluated aesthetic scores ranging from 1,085 to 1,910, which was used as the learning dataset for a CNN (Fig 1A).

## Predicting the aesthetic values with the CNN

We trained a CNN to estimate the aesthetic value of new fish images and eventually predict values for the remaining 4,400 images in our collection (Fig 1A). We used a ResNet50 model pretrained on ImageNet [32], a popular CNN used for image classification. We replaced the last classification layer by a regression layer, and performed a 5-fold cross-validation to fine-tune this layer and the last convolutional block (see Methods and Text E in S1 File). The $r^2$ of the linear regression between the values predicted by the CNN and the evaluated values from the validation set (averaged over the 5 folds of the cross-validation) was 0.79 ± SD 0.04 (Fig 1B). Applying the trained model to the 4,400 unevaluated fish images, the predicted values ranged from 1,153 to 1,980 (Fig 1B and 1C). The remaining analyses of our study were based on the values of 4,881 images: the 481 evaluated during the online surveys and the 4,400 with predictions from the CNN. We used several images per species to account for intraspecific morphological differences and used the highest predicted value for each species (S2 Data), thereby assuming that humans tend to focus on the most attractive representation of a species (see Discussion).

## Determinants of aesthetic value

For each of the 4,881 images in our collection, we extracted 17 image features potentially linked to the aesthetic value, regrouped into 4 classes: (a) the color heterogeneity; (b) the geometry of color patterns; (c) the perceptual lightness and saturation; and (d) the shape of the fish outline (see Methods and Text A in S1 File). After eliminating nonsignificant features using a backward selection procedure, we ended up with 9 features that explained a substantial amount of variation in aesthetic value with a linear model ($r^2$ = 0.64, $p$-value < 0.001, Fig 2A, Text F in S1 File). The most significant features were color heterogeneity, color saturation, and elongatedness (see Fig 2A; Text A, F, and Fig D in S1 File). The projection of fish aesthetic values on the 2 first axes of the principal component analysis (PCA) performed on the selected features (Fig 2B) confirms that fishes with the highest aesthetic values are those with high color heterogeneity and saturation as well as more circular body shapes (low elongatedness). Fishes with a high standard deviation in perceptual lightness and presence of several repeated patterns also have high aesthetic values. At the opposite of the gradient, drab fishes with elongated body shape and no clearly delineated color patterns have low aesthetic values (Fig 2B).

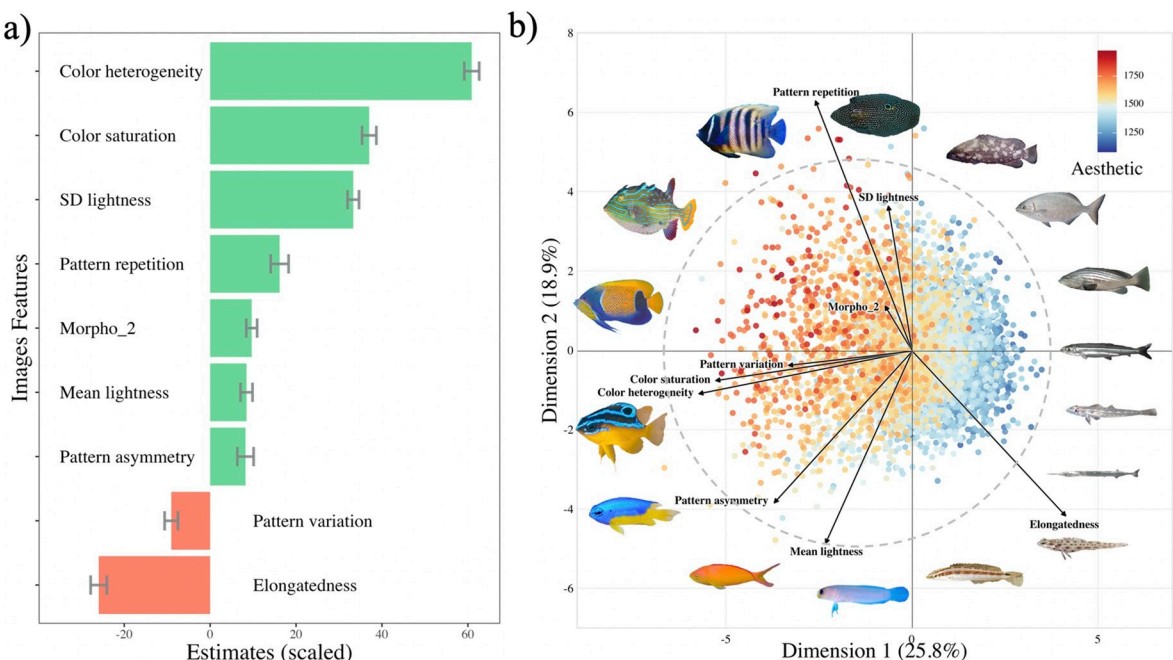

**Fig 2. Image features analysis.** (a) Regression coefficients (with standard errors) from the final model between the aesthetic value and the 9 significant image features (see Text A in S1 File for a complete description of the image features). The variables have been scaled to visualize how the magnitude of the effects differs between them. Most important variables were: color heterogeneity, color saturation, standard deviation in lightness (SD lightness), pattern repetition, and body elongatedness. (b) Principal Component Analysis (PC1 and PC2) performed with the 9 significant images features (see Fig 2A, Text A, F in S1 File for a description of the image features). Points (fishes) are colored by their aesthetic values, and image feature vectors are projected on the 2 axes. Examples of fishes (chosen on the perimeter of the distribution) are provided for illustration. Clockwise order: *Calloplesiops altivelis, Epinephelus ongus, Kyphosus vaigiensis, Epinephelus costae, Jenkinsia lamprotaenia, Phyllogobius platycephalops, Belone belone, Ctenogobiops crocineus, Suezichthys devisi, Opistognathus aurifrons, Pseudanthias ignitus, Pomacentrus auriventris, Mecaenichthys immaculatus, Pomacanthus navarchus, Aracana aurita, Pomacanthus sexstriatus*. See S1 Data for image copyright. Data and code required to generate this Figure can be found in https://github.com/nmouquet/RLS_AESTHE.

## Phylogenetic signal in aesthetic value

We then explored the link between fish aesthetic value and evolutionary history [33–35]. The mean age (over 100 trees) of species ranged from 0.41 to 165.53 My. We found that the youngest species have the highest aesthetic value (Fig 3A) and a significant negative relationship between the aesthetic value and the mean Evolutionary Distinctiveness (Fig N in S1 File): species with long, isolated branches in the phylogenetic tree tend to have lower aesthetic value than less phylogenetically isolated species. These results identify members of the most recently diversified families as aesthetic hotspots in the tree (Fig 4, Text G and Fig O in S1 File). For example, the families *Pomacanthidae* and *Acanthuridae*, respectively, have a mean aesthetic value of 1,719 ± SD 176 (*n* = 42 species) and 1,590 ± SD 150 (*n* = 67), whereas the oldest families *Scombridae* and *Carangidae* have respective mean aesthetic values of 1,228 ± SD 54, (*n* = 18) and 1,278 ± SD 86 (*n* = 49). Pagel's λ estimated on the entire tree confirms this strong phylogenetic signal (λ = 0.74 ± SD 0.01, *p*-value < 0.001). At a finer phylogenetic resolution, we also found significant phylogenetic signals within 9 families: *Acanthuridae, Balistidae, Carangidae, Chaetodontidae, Haemulidae, Holocentridae, Pomacanthidae, Sciaenidae*, and *Scorpaenidae* (λ ≥ 0.50, *p*-values < 0.05, Text G and Table B in S1 File), indicating that the clustering of aesthetic value also operates at the genus level within some families.

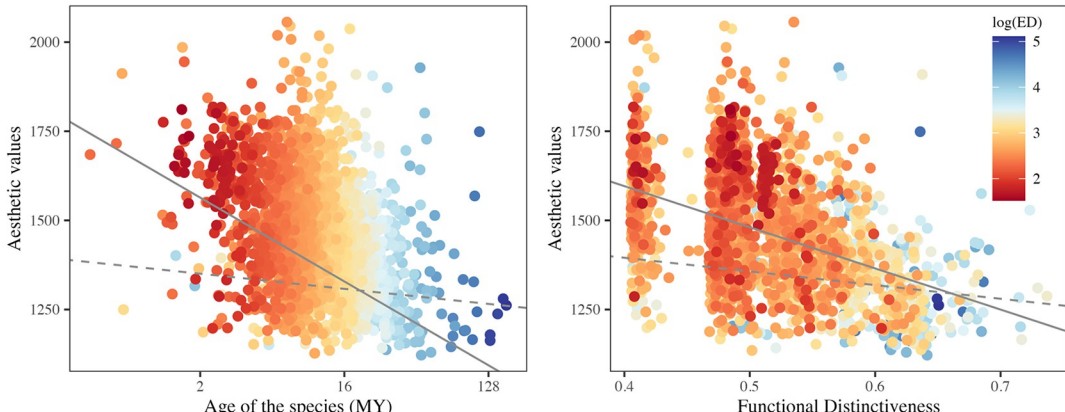

**Fig 3. Phylogenetic history and ecological originality.** (a) Relationship between the aesthetic value and the age of the species (log transformed) in millions of years (averaged over 100 trees) without (plain line) and after (dashed line) accounting for phylogenetic relatedness. Both models show significant negative slopes (considering phylogenetic relatedness: slope = −78.4, *p*-value < 0.001; not considering phylogenetic relatedness: slope = −14.1 ± SD 1.7, *p*-value < 0.005, over the 100 random trees). (b) Relationship between the aesthetic value and the functional distinctiveness of species without (plain line) and after (dashed line) accounting for phylogenetic relatedness. Both models show significant negative slopes (considering phylogenetic relatedness: slope = −1,154.2, *p*-value < 0.001; not considering phylogenetic relatedness: slope = −383.3 ± SD 26.5, *p*-value < 0.001, over the 100 random trees). On both panels, species' Evolutionary Distinctiveness (averaged over 100 trees and log transformed) have been used to color the points from low (dark red) to high (dark blue) values. Data and code required to generate this Figure can be found in https://github.com/nmouquet/RLS_AESTHE.

## Aesthetics value across ecological trait space

To characterize ecological originality of fish species (hereafter called functional distinctiveness), we used 8 ecological traits that describe body size, diet, behavior, and habitat use [37]. We computed Gower's pairwise distances between species to estimate their functional distinctiveness [38], and found that the most unique species, with respect to their trait combination, have lower aesthetic values than more ecologically redundant species (Fig 3B). A closer look at the distribution of aesthetic values within each trait category (Text H and Fig P in S1 File) shows that most attractive fishes are associated with hard substrates (as opposed to sandy patches within or along the edges of the reef), are demersal (hover above but near the bottom), active during the day, feed on corals or by excavating the reef surface, of intermediate body size, and prefer warmer waters. The least attractive fishes tend to be pelagic, nocturnal, eat other fishes or plankton, have either small or large body size, and prefer cooler waters.

## Conservation status and aesthetic value

We categorized fish IUCN status into 3 groups: 190 species in our dataset are Threatened (TH: Critically Endangered, Endangered, or Vulnerable), 1,602 species are Least Concern (LC: Least Concern or Near Threatened), and 556 are Not Evaluated (NE). We found significant differences between the mean aesthetic value of species in these 3 groups (1-way ANOVA, *p*-value < 0.001). Tukey's post hoc tests show that Threatened fishes have the lowest aesthetic values on average, and Least Concern fishes the highest (Fig 5A), although variability within these groups was high. Further, we grouped the Least Concern and Threatened fishes into an Evaluated category and found that they had a higher mean aesthetic value than the Not Evaluated category (1-way ANOVA, *p*-value < 0.001).

We also compared species aesthetic value with their importance to fisheries. According to FishBase, 594 of our species are classified as "Non commercial," 83 as "Subsistence fisheries," 368 as "Commercial," and 43 as "Highly commercial." The remaining 1,329 species are "Data

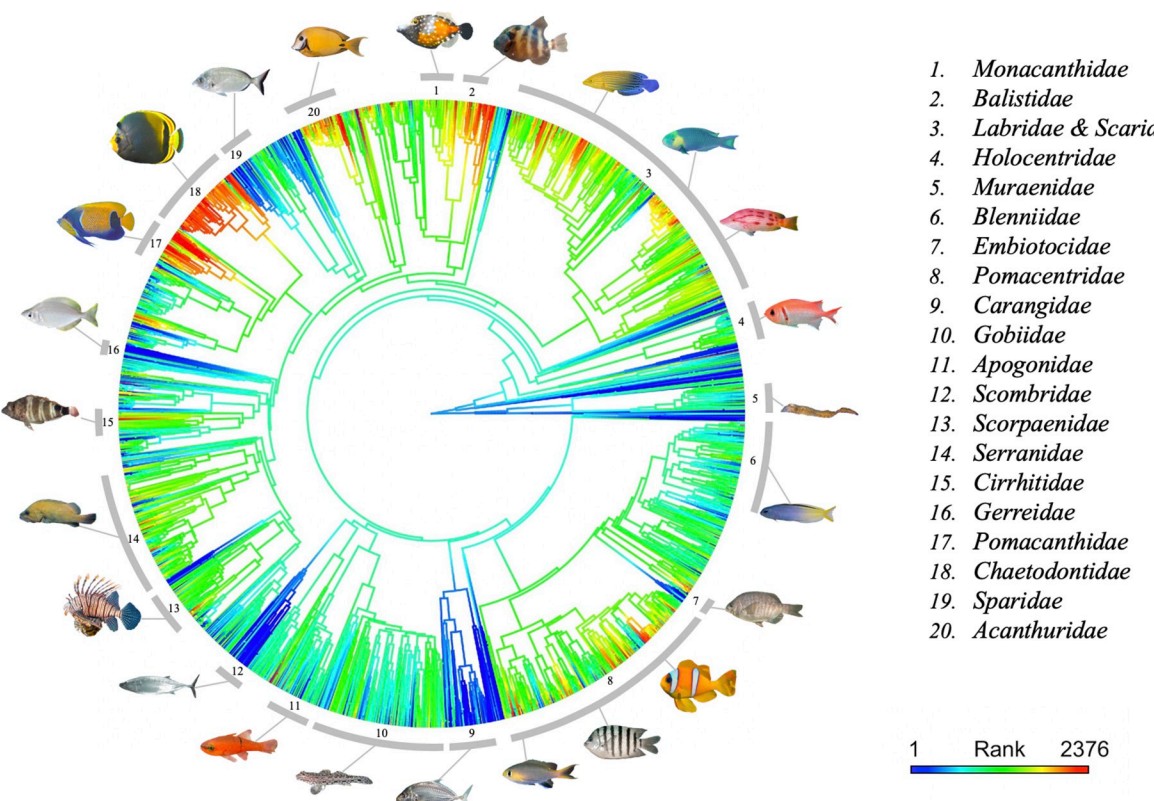

1.  *Monacanthidae*
2.  *Balistidae*
3.  *Labridae & Scaridae*
4.  *Holocentridae*
5.  *Muraenidae*
6.  *Blenniidae*
7.  *Embiotocidae*
8.  *Pomacentridae*
9.  *Carangidae*
10. *Gobiidae*
11. *Apogonidae*
12. *Scombridae*
13. *Scorpaenidae*
14. *Serranidae*
15. *Cirrhitidae*
16. *Gerreidae*
17. *Pomacanthidae*
18. *Chaetodontidae*
19. *Sparidae*
20. *Acanthuridae*

**Fig 4. Aesthetic values across the tree of life.** Phylogenetic tree of the 2,417 fishes. Aesthetic values are mapped over the entire phylogeny with a color gradient obtained by estimating states at internal nodes with maximum likelihood [36]. For illustration, we have highlighted 20 families with contrasted aesthetic values using gray arcs and show examples of fishes for each family. Clockwise order: *Cantherhines macrocerus, Pseudobalistes naufragium, Anampses femininus, Scarus spinus, Bodianus unimaculatus, Myripristis jacobus, Gymnothorax annasona, Meiacanthus atrodorsalis, Embiotoca jacksoni, Amphiprion bicinctus, Abudefduf bengalensis, Chromis alpha, Carangoides chrysophrys, Istigobius decoratus, Apogon pacificus, Sarda australis, Pterois miles, Acanthistius ocellatus, Amblycirrhitus pinos, Parequula melbournensis, Pomacanthus navarchus, Chaetodon flavirostris, Diplodus puntazzo, Acanthurus tristis*. See S1 Data for image copyright. Data and code required to generate the phylogenetic tree can be found in https://github.com/nmouquet/RLS_AESTHE.

deficient." One-way ANOVA (*p*-values $< 0.001$) and Tukey's post hoc tests (all *p*-values $< 0.002$) showed that species for which no data is available, species of no fishery interest, and species important for subsistence fisheries have similar aesthetic value. The differences between the mean aesthetic value of the other categories are statistically significant, with the Highly commercial species having the lower aesthetic values (Fig 5B).

## Discussion

Our study provides a global picture of variation in aesthetic value, an important but under-studied facet of biodiversity, of reef fishes. It reveals some predictable differences among species and potential mismatches with conservation priorities. The aesthetic values of reef fishes are highly heterogeneous, with the most beautiful fishes being tightly packed into small regions of both the phylogenetic tree and the ecological trait space of the world's reef fish fauna. In contrast, the most ecologically and evolutionary distinct species and those recognized as threatened tend to be considered less attractive.

The set of 481 images evaluated through our online survey allowed us to train a deep learning algorithm to predict the aesthetic value of 4,400 fish images without having to arbitrarily predefine any visual features. The predictive power of our algorithm ($r^2 = 0.79$) was high

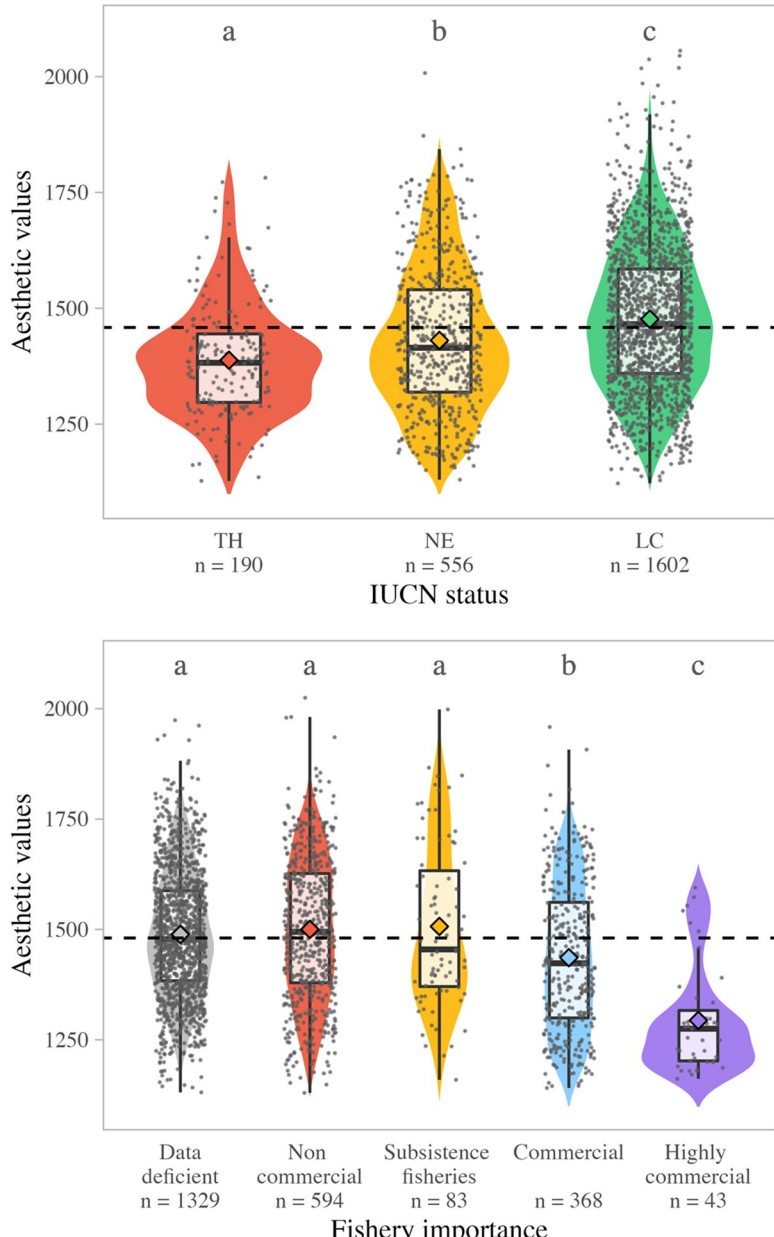

**Fig 5. Fish aesthetic value and conservation status.** (a) Violin plot of the aesthetic value of reef fishes for three groups of conservation status: TH, NE, and LC. Letters indicate significant differences between the groups (Tukey *p*-values are respectively *p* < 0.01 between TH and NE, *p* < 0.001 between LC and TH, and *p* < 0.001 between LC and NE). (b) Violin plot of the aesthetic value of reef fishes for the 5 groups of fishery importance: "Data deficient" = no data available; "Non commercial" = no interest for fisheries or potential interest or minor interest; "Subsistence fisheries" = importance for subsistence fisheries; "Commercial" = commercial importance for fisheries; "Highly commercial" = high commercial importance for fisheries. Letters indicate significant differences between the groups (all Tukey *p*-values are < 0.002). Data and code required to generate this Figure can be found in https://github.com/nmouquet/RLS_AESTHE. LC, Least Concern; NE, Not Evaluated; TH, Threatened.

despite the relatively small size of our learning dataset and the cross-validation. This is largely due to the transfer learning procedure that takes advantage of pretrained CNNs and where only the weights of the last few layers needed to be specifically tuned to the new task. The high

predictive power can also without doubt be explained by our image preprocessing, including background removal, and position and size standardization. This preprocessing required a considerable amount of work but could be automated in the future by a computerized segmentation and alignment of fishes [39], allowing for a broader use of images and/or videos from larger collections, including from the internet or social media. Our study highlights multiple mechanisms by which recent advances in computer science and artificial intelligence can be used by ecological studies to improve sample sizes. Beyond the scope of our study, our deep learning algorithm could be easily adapted to fishes from other ecosystems such as those from rivers and lakes, and even to other taxa such as birds, mammals, reptiles, or amphibians, for which millions of images are now available online. Our CNN could provide a valuable tool for studying these other taxa in 2 ways: as a pretrained network to fine-tune through transfer learning or without retraining to investigate the generalization of human aesthetic judgments from one animal taxon to others.

As we did not need to predefine any visual features for the automation of image scoring, we could then use post-analysis of image features to characterize the visual components contributing most to the aesthetic value of reef fishes. Our linear model combining 9 features potentially linked to aesthetic value (see Text A and F in S1 File) explained a substantial amount of the variation in aesthetic value ($r^2 = 0.64$, Fig 2A). The features that explained the highest aesthetic values were heterogeneity in color and lightness, saturation of colors, the presence of well delineated and repeated patterns, and the circularity (versus elongatedness) of the body shape. These results can be interpreted through the lens of neuro-aesthetics, which relates aesthetic properties to the activation of dopamine neurotransmitter systems within the human brain [40], and to the processing fluency theory, which hypothesizes that aesthetic value is determined by the dynamics of information processing in the brain, and that stimuli that are easy to process are judged beautiful [41]. For example, components that can easily be separated from the background, or visual features that can obviously be grouped into recognizable objects, both trigger pleasure and tend to be judged beautiful [42]. Some of the features explaining aesthetic value in reef fishes could similarly be explained by fluency. For example, high color heterogeneity and well-delineated patches of contrasted lightness, as observed in angelfishes (*Pomacanthidae*) and butterflyfishes (*Chaetodontidae*), increase fluency by facilitating detection and recognizability [43]. The elevated aesthetic value of fishes with circular body shape echoes a general preference of humans for objects with curved outlines [44,45], which can also be explained by fluency given that curved lines are more predictable and thus more easily encoded by the visual system than angular lines [41]. Overall, our analysis suggests that predictable visual features explain human preferences for fishes, creating a strong bottleneck in the diversity of species that are likely to be considered as beautiful by the general public.

This aesthetic bias toward particular color and morphological features leads to strong clustering of aesthetic value across the Tree of Life and ecological trait space. Some families show overall high aesthetic value, such as butterflyfishes (*Chaetodontidae*), while some others show overall low aesthetic value, such as jacks and pompanos (*Carangidae*). This strong phylogenetic signal is probably explained by the relatively high ecological niche conservatism within fish families. For example, most *Chaetodontidae* are found among living corals, and most *Carangidae* live in open water or above the reef. The ecological niche plays a prominent role in determining color patterns, through its effect on social behavior, feeding activities, and most notably camouflage [46]. We thus suggest that low trophic level species living on rocks and coral have higher aesthetic values than higher trophic level species, such as piscivores, because of their need to blend in with more color-rich habitats (to our trichromatic color vision and under white light) for camouflage purposes [47]. These rock and coral habitats are also visually complex, being highly fractal [48]. By adapting their visual patterns to these structured

habitats, coral reef fishes would thus "benefit" from a preference bias that we humans have for fractal patterns [49]. On the other hand, pelagic fishes that have to hide within a poorly contrasted and homogeneous habitat would be considered less attractive [50].

The ecological niche could also influence our perception of fish beauty through its effect on locomotion. Most pelagic fishes are good swimmers with a fusiform body shape that reduces frictional forces, but also increases the angularity of body outline, which appears to be associated with reduced aesthetic value. Conversely, some demersal species that rely on pectoral fins to swim [51] have a circular body shape, associated with higher aesthetic value. We also found that younger species (in evolutionary history) and more diversified clades show higher aesthetic value (Fig 3A), which could be explained by the relatively "recent" diversification of particular families associated with the exploitation of coral reef niches. For example, among the families with high aesthetic values, the angelfishes (*Pomacanthidae*) appeared during the Eocene, with all genera in place by the mid-Miocene [52]. Likewise, the butterflyfishes (*Chaetodontidae*) moved to coral reefs in the Miocene, with subsequent cladogenesis [53], the origin of the triggerfishes (*Balistidae*) is dated to late Miocene [54], and the boxfishes (*Aracanidae*) appeared during the Pliocene [55].

Together, these results highlight how our aesthetic judgment, which is linked to the ability of the human brain to process visual information, could generate strong bias in our emotional engagement with reef fishes. This bias of perception was suggested by a previous study performed on 116 coral reef fishes, which found that the least attractive fishes have a higher functional richness than the most attractive ones [16]. Our study extends these results to 2,417 reef fish species and shows that the distribution of aesthetic values is strongly skewed along both ecological and evolutionary dimensions. The fact that human visual aesthetic preferences are predictable and narrower than the diversity of reef fish life forms is not surprising in itself, but has important implications as our emotional perception of nature is one of the major drivers of our interest in it. The aesthetic bias could influence decisions in reef fish research and conservation. For example, Bellwood and colleagues [18] have previously shown that scientific published literature is biased towards yellow fishes which, given our results, are likely to be among the most attractive ones. Likewise, in mammals, the most beautiful species are subject to more research effort than less attractive species [13]. Our study extends these results to reef fish conservation as we found a mismatch between the aesthetic value of reef fish species and their conservation status. Less attractive fishes are also less well represented in assessments for the IUCN Red List, and are thus paying what could be considered as a form of "aesthetic-related debt" in a time where the amount of effort devoted to conservation is limited. This debt is a major concern given that, as shown here, the most threatened species (as evaluated for the IUCN Red List) are those with the lowest aesthetic value (Fig 5A). We also found that the less attractive fishes are also more at risk of overexploitation by fisheries (Fig 5B). Beyond species, this debt could have consequences at the whole reef ecosystem level, as we found that the less attractive species have the highest ecological distinctiveness, and thus provide the highest diversity of ecological functions. We thus suggest that the elevated extinction risk of the less attractive fish species might also have disproportionate and overlooked effects on reef ecosystem functioning.

Our approach has some limitations and considerations that may affect the accuracy of individual values, including the use of maximum predicted aesthetic values for each species and the pretreatment of images shown to the human respondents. Some reef fishes show extreme polymorphism in color or body shape (for example, male versus female and adults versus juveniles), but the high number of species included in our study prevented the gathering of an exhaustive collection of all the existing morphs. Because we aimed at providing the most generalizable results to the world's reef fishes, we deliberately chose to maximize the number of

species rather than finely characterizing intraspecific variation. When several images were estimated for a species, we kept only the maximum aesthetic value to characterize this species. This was motivated by our desire to account for a possible "halo effect," a cognitive tendency of people to generalize the evaluation of the most valued item to other, less valued item of the same category [56]. Nevertheless, reproducing the analysis using averaged, rather than maximum, aesthetic values by species yielded similar results (Text I in S1 File). We further acknowledge that background composition in images could also be important, as it has been shown to influence the perception of beauty in psychological tests [57]. Yet, inclusion of the background would have made it uncertain how much the values related to the fish versus the background. To maximize comparability, we also scaled all fishes to the same size (500 × 500 pixel squares), but if represented proportionally to real size, bigger species may have had an "aesthetic bonus." People tend to prefer bigger individuals, independently of their intrinsic aesthetic value [58], as it has been shown for birds [59]. This size issue has also motivated our decision to focus only on ray-finned fishes and exclude bigger fishes such as sharks, for which the size aesthetic bonus and charismatic nature make them attractive to people, even though their body shape and coloration would suggest lower aesthetic value. Finally, we removed the Pleuronectiformes (14 species) and Syngnathiformes (31 species) to help standardize the morphologies of fishes within our dataset. Within the Syngnathiformes, seahorses might be considered as highly attractive, yet possess distinct functional traits; a combination which would contradict one of our main findings. They represent, however, only 14 species in the RLS dataset of most common reef fishes encountered by survey divers used for this analysis, which would not likely change the overall trends or tendencies we report. On the other hand, the Pleuronectiformes would be more likely to be scored as less attractive, and hold original ecological traits; a combination which would have reinforced our results. All these limitations must be kept in mind when interpreting our results, but we consider them unlikely to affect the generality of our study's conclusions for conservation and ecology. We rather see them as future research opportunities and believe that the extent of our study (2,417 species), the large taxonomic coverage (139 families of reef fishes), and the range of predicted aesthetic value (Fig 1B) provide solid generality and robustness to our main findings.

We also anticipated that the results of our predictive model could be biased by respondents individual factors, given that aesthetic preferences can vary with age, gender [60], social group [61], or culture [62]. Nevertheless, we found no such effect when analyzing the robustness of the match outcomes (probability for an image to win a match) to geographic or sociocultural backgrounds. However, the absence of sociocultural effects should be interpreted with caution because of the relatively limited diversity of sociocultural backgrounds among our respondents and geographical biases. For example, 62.4% of the respondents were French, 84.9% had a Bachelor's degree or a higher diploma, and 52% had some experience scuba diving (Text C in S1 File). Despite these biases, our results highlight some degree of universality in the aesthetic values of fishes that probably outweigh sociocultural differences. This generality may be explained by our protocol for evaluating the aesthetic value. By presenting images by pairs and asking people to simply choose the image they found the most beautiful, we ought to force people to make rapid choices based on a bottom-up, stimulus-driven processing of visual information rather than a cognitive-driven evaluation based on previous knowledge [63]. Although we believe our general conclusions should hold, we would presumably have found differences in individual scores had we compared respondents spanning a much larger breadth of cultural, social, or demographic backgrounds, or asked questions that would have triggered more cognitive processing. Addressing this would require a more balanced pool of respondents and opens interesting perspectives for future research in the framework proposed by "personalized ecology" [64].

Overall, our results show that a nonmaterial facet of NCP, aesthetic value, mismatches with the material and regulating NCP facets provided by fishes. This mismatch strengthens the case for recognizing multiple aspects of NCP, but paradoxically, it also implies the potential for an aesthetic driven debt if the less attractive fishes are more exploited and receive less conservation/research efforts. This should motivate strengthened research efforts in evaluating the aesthetic value of biodiversity more generally, and the extent to which perceptual and emotional biases translate into biases in research, awareness, and conservation. Capitalizing on the capacity of artificial intelligence to work on extensive datasets, future research should also be able to expand our aesthetic valuation at higher levels of biological organization, such as assemblages [65] and whole ecosystems. Investigating the common patterns in aesthetic values across taxa such as birds, mammals, reptiles, or amphibians would also be an exciting research avenue, and could serve to fuel research and conservation programs. Our study further revealed a high consensus across participants in the aesthetic evaluation, although we acknowledge this could change through time, education, and with connectedness to nature [5,66] (even if not captured in our study through an influence of the sociocultural background of participants). We do not believe that identified aesthetic biases will always turn into debts. The first step in minimizing the impacts of these biases for conservation success will be better communication to the public, policy-makers, conservation NGOs, and researchers on the links between the nonmaterial components of nature's contribution to people [1] (here aesthetic value), the ecology of species and the roles they perform in ecosystems [5].

## Methods

Most analyses were carried out using R *v.3.6.0* (specific functions within specific packages are indicated in italic). All relevant code and data are available from the associated GitHub repository (see sections Data and Code availability).

### List of species

Fishes are the most diverse class of vertebrates, with more than 30,000 species and a huge diversity of shapes, sizes, and colors [67]. We sought to sample this diversity in an ecologically and socially meaningful subset, focusing on reef fishes. We used the RLS database [29] to identify a subset that could be representative of those species likely to be directly encountered and seen by people, but still spanning all major ocean basins and reef areas. The RLS database only includes records from standardized, quantitative visual surveys by scuba divers on rocky and coral reef habitats in shallow waters (mostly 0 to 20 m depth). It is the largest single-method dataset available for reef fishes (134,759 abundance records from 1,879 sites, representing 2,397 species) and has an associated trait database [37]. We decided to focus only on ray-finned fishes (*Actinopterygii*), and excluded from this clade the orders *Pleuronectiformes* (14 species) and *Syngnathiformes* (31 species) to remove the very unusual morphologies that would have been harder to present in the standardized questionnaire and to characterize further in our image features analysis (see below). As one of our foci was on IUCN status, we supplemented the remaining 2,280 RLS fish species list with 137 reef-associated species identified from FishBase that are classified as threatened (critically endangered, endangered, vulnerable), but not observed in the RLS database, so that our final list covered most of the threatened fishes listed on the IUCN red list that are also considered "reef-associated" (according to FishBase). We ended up with a list of 2,417 reef fish species belonging to 139 families (S2 Data). Taxonomic names were checked using the World Register of Marine Species database WoRMS [68,69].

## Photographic material

Images were collected on the internet, focusing first on particular sources considered as reliable for species identity. Most of the material (3,198 images) was collected from RLS "Reef Species of the World" web pages (https://reeflifesurvey.com/species/search.php), FishBase (https://www.fishbase.se/), EOL (https://eol.org), and Fishes of Australia (https://fishesofaustralia.net.au/). We also collected material from Google Images (1,683 images; see Text B in S1 File and S4 Data), but only when the species could be unambiguously identified. When available, several images were collected for species with color polymorphism, or with morphological differences between males and females, or between adults and juveniles. The final dataset included 4,881 images. Note that for 8 species we could not find images and we removed them from our analysis (see S2 Data). For each image, we removed the natural background using Clipping Magic (https://fr.clippingmagic.com/) and standardized the position of the fish with photoshop. Correction of color saturation (in the blue and the green essentially) and luminosity was also performed when needed using Photoshop. Final images represent a single individual, horizontally aligned (mouth on the left, tail on the right), and displayed against a white background. All images were resized to 500 × 500 pixels at 96 dpi (Fig 1A).

## Direct evaluation of fish aesthetic values

Our first objective was to build the learning dataset for our deep learning algorithm. That is, a dataset of images that are representative of the range in different visual features found among the 4,881 images of our whole collection, and for which individual aesthetic values would be evaluated by the public. To ensure the robustness of the method, the surveyed photographic material must be representative of the taxa considered and evaluated by a sufficiently large panel of humans with contrasted sociocultural background, which can be achieved by using online surveys [14,16].

Our learning dataset was obtained from combining results from a previous study where we empirically evaluated the aesthetic values of 157 images of reef fish species [16] and a new online evaluation of 324 images. The 324 images were subsampled from our 4,881 images collection to maximize variation within visual features. For doing so, we performed a PCA of the visual features measured on all of the 4,881 images (see below and Text A and B in S1 File). The 5 first dimensions of the PCA (representing 77% of the total explained variance) were used to construct a convex 5 dimensional volume. The 162 images were then randomly drawn near the vertices of this volume, and the other 162 images were randomly drawn within the remaining volume. We added 21 further images from the 157 images previously evaluated [16] so that we could quantify any differences between the 2 surveys (see below) and merge the 2 datasets. The robustness of our procedure can be visualized on Fig E in S1 File (see also Text B in S1 File), which shows the high overlap between the 2 ellipses (on the first 2 axes of the PCA) representing the 99% confidence intervals of both the 345 selected images and the whole image set (see Fig E in S1 File).

The public online survey included 2 sections: (a) the aesthetic questionnaire itself during which each participant (hereafter called "respondent") had to choose the image they found the most beautiful for 30 pairs (hereafter called "matches") randomly sampled without replacement among the 345 images; and (b) a questionnaire (Text C in S1 File) to gather information on the sociocultural background of respondents (gender, age, education, experience with diving and spearfishing, fishkeeping, place of living, distance from the sea, exposure to natural space, and knowledge about coral reef fishes). The survey was anonymous and available in French and English to the public on a dedicated website (https://www.biodiful.org/) between February and June 2019. The website contained an introduction that stated the objective of the

research and an ethical statement to guarantee anonymity to the respondents (see Text C in S1 File). The survey was distributed through massive emailing to authors' contacts, various mailing lists in France and abroad (scientific societies, universities, aquariums, NGOs, etc.) and social media (Facebook and Twitter) asking the respondents to share the survey to their families and friends. The answers of 13,000 respondents without color perception issues were pooled and we computed the aesthetic values of the 345 images with the Elo algorithm [30] using the *EloChoice v0.29.4* R package [70] with 1,000 bootstrapings (Text D in S1 File).

Finally, we computed linear regression between the aesthetic values of the 21 images common to the previous [16] and the new online surveys, and used the intercept and slope of this regression to merge the 2 datasets (Text D in S1 File). The final set used to train the deep learning algorithm thus included 481 images with continuous aesthetic values.

## Predictive model of fish aesthetic values

To predict the aesthetic value of the whole set of images of our collection, we used a deep learning algorithm trained with the 481 evaluated images. Deep learning algorithms are most popular for their application to identification or classification tasks, but they are also suited for predicting continuous variables [71]. For computer vision tasks like the regression between the information of an image and a continuous variable, CNNs are considered to perform best [71]. We applied a transfer learning procedure by fine-tuning a ResNet model, one of the most popular architectures in artificial intelligence [72], pretrained on ImageNet [32]. Preliminary tests showed that the commonly used image size of $224 \times 224$ pixels provided the best results (Text E in S1 File).

We tested 2 different architectures, ResNet18 and ResNet50, which differ by the number of layers and thus the number of learnable parameters. After the hyperparameter configuration was optimized for both architectures, ResNet50 performed best and was thus retained as the predictive model. We performed 5-folds cross-validation such that the learning dataset of 481 images was divided into 5 parts of similar size with all the images of a given species in the same part. ResNet50 was thus trained on 4 parts (training set) and evaluated on the fifth one (evaluation set), repeating the procedure 5 times (each time changing the evaluation set). To limit the risk of overfitting, the training set was artificially augmented at each iteration by applying random rotation (−5; 5 degrees). We used the r2 of the linear regression between the values predicted by the model and the evaluated values of the learning set to estimate the accuracy of the model. Further details on the parameters of the model and the learning procedure can be found in Text E in S1 File. Once the accuracy of the model on the 5 folds of the cross-validation was satisfying, we trained the model one last time on the 481 images and used the weights of this model to predict the aesthetic values of the remaining 4,400 unevaluated images. Data augmentation, fine-tuning of the models, and prediction of the aesthetic values were carried out using Python 3.7, Pytorch 1.4.0, and torchvision 0.5.0.

As some species with intraspecific morphological differences were represented by several images in our collection (see section "Photographic material"), we decided to characterize species level aesthetic with the highest aesthetic values obtained among images of the same species.

## Visual features analysis

The choice of the visual features analyzed in the postprediction analysis was based on literature review of previous works which studied the aesthetic value of biodiversity (Text A in S1 File). We chose 4 different classes of features: (a) the color heterogeneity; (b) the geometry of color patterns; (c) the perceptual lightness and saturation; and (d) the shape of the fish body outline.

The heterogeneity of colors and the geometry of color patterns were measured after using a K-means clustering algorithm to separate colors in the CIELAB color space. The lightness, that is, how close to white (high values) or black (low values) a color is, and color saturation were measured using the HSV color space. Features describing the fish shape were computed using morphometric analysis relying on elliptical Fourier transformation. Altogether, we obtained 17 different visual features (see Text A in S1 File for a complete description). We estimated the individual contribution of each visual feature to the aesthetic value obtained for the 4,881 images using a multiple regression approach (see section Statistical analysis). A PCA was performed with the scaled selected features (*dudi.pca* function in the *ape v.1.7–16* R package), and the image coordinates (colored by their aesthetic values) were projected on the first 2 axes of the PCA.

## Evolutionary history

We extracted the taxonomy of the 2,417 species from the WoRMS database via the *taxize v0.9.91.91* R package [73]. We then used the *fishtree v0.3.2* R package [33, 74] to compute the phylogenetic tree. This tree includes grafted species for which the genetic information is not directly available, but which are known from other published phylogenies or inferred from taxonomic positions. Hence, more than one branch descends from a single node in some parts of the tree, whereas a phylogenetic tree should have dichotomous divisions from common ancestors. To bypass these polytomies, 100 realizations of a stochastic polytomy resolver placing missing speciation events were used [75]. We extracted the age of the species as the length of the branches from the first node of the tree to the species' leaf. We tested for a phylogenetic signal of the aesthetic values with Pagel's λ coefficient [35] (see section Statistical analysis). Finally, we computed the Evolutionary Distinctiveness (ED; which reflects the phylogenetic isolation of a given species [34] of each species using the function *evol.distinct* from the *picante v1.8.1* R package [76]. ED is high when the species is phylogenetically isolated, that is, it has a long unshared branch in the phylogenetic tree. The 3 indices, species age, Pagel's λ, and ED were computed on the 100 randomly resolved trees and averaged.

## Ecological traits

We used the RLS trait database [37] that covers body size (maximum length), feeding ecology (trophic group, trophic breadth), behavior (water column position, diel activity pattern), and habitat use (preferred temperature, habitat complexity). See Text H in S1 File for details. A total of 129 species had more than 50% missing trait values, while 107 species had less than 50% missing traits, and we used the R package *missForest v1.4* [77] to impute them (Text H in S1 File) to complete all ecological traits for 2,288 species (94.7% of the dataset). To characterize species ecological originality, we computed functional distinctiveness ($D_i$), which measures the average functional distance of a focal species to the other species [38]. We first computed a matrix of Gower distances of normalized species ecological traits using the function *dist.ktab* of the *ade4 v1.7–13* R package [78] and then used the *funrar v1.4.1* R package [79] to compute functional distinctiveness. Aesthetic values were also compared among categories or along values of ecological continuous traits (Text H in S1 File).

## Conservation status and importance for fisheries

We used the *rfishbase v3.1.1* R package [80] to obtain the updated IUCN status of the 2,417 fishes [81] and their importance for fisheries. To simplify interpretation, we grouped species into 3 categories according to their IUCN status: "Threatened" (TH) refers to Critically Endangered, Endangered, and Vulnerable species; "Least Concern" (LC) refers to Least Concern and

Near Threatened species and "Not Evaluated" (NE) refers to Not Evaluated species. The 69 species had Data Deficient status and were removed from the analysis. We also grouped species into 5 categories according to their importance for fisheries: "Highly commercial," "Commercial," "Subsistence fisheries," "Non commercial," and "Data deficient." The "Non commercial" category refers to species indicated as "Minor commercial," "Of no interest," or "Of potential interest."

## Statistical analysis

To test for the robustness of the matches' outcome (probability for an image to win or not a match) to sociocultural background, we computed a generalized linear mixed model (GLMM) with a binomial error structure (using the *glmer* function from the *lme4 v 1.1–26* R package) in which the image was considered as a random effect variable to order the sociocultural variables according to their individual effect on the response variable (see also Text C in S1 File).

To estimate the individual contribution of each visual feature to the aesthetic value obtained for the 4,881 images, we used a multiple regression approach (see Text F in S1 File for a complete description). We first computed a correlation matrix between all features (using Pearson correlation coefficients): when 2 or more features were correlated (threshold $r < 0.7$), we kept only the feature with the highest correlation with the aesthetic value. We then created a linear model (with Gaussian response) explaining aesthetic values where each feature was ordered in the model according to its independent contribution to the total variation in the response variable. We eliminated non-significant terms using a backwards selection procedure, to derive a minimal adequate model and used the coefficients (scaled) of the final model to measure the contribution of each selected feature to the aesthetic value.

To test for a phylogenetic signal of the aesthetic values, we used Pagel's λ coefficient [35]. Pagel's λ characterizes the relation between the similarity of a given trait (here the aesthetic value) and the phylogenetic distance between species. It represents the possibility to reconstruct the tree with the studied trait only. A null λ leads to a single polytomy for the basal node while a value of 1 gives the exact tree. A *p*-value is computed by randomizing the data in order to identify the families for which an inner signal is detected [82]. This analysis was undertaken on the entire tree and only for the families for which we had more than 5 species in our dataset. We used the *phylosig* function of the *phytools v 1.0–1* R package.

To test the relationship between aesthetic values with both the age of the species (in millions of years) and the functional distinctiveness, we used both linear models and linear models accounting for phylogenetic relatedness using the *phylolm* function of the *phylolm v 2.6.2* R package over the distribution of 100 resolved phylogenetic trees (the 100 *p*-values were combined using the *hmp.stat* function of the *harmonicmeanp v 3.0* R package).

To compare aesthetic values respectively across the 3 IUCN categories and the fisheries importance categories, we performed a 1-way ANOVA and Tukey multiple pairwise comparisons using the *aov* and *TukeyHSD* functions of the base-attached *stats* R package.

## Data and materials availability

Most of the data used in this paper are freely available and downloadable from the web. Data on IUCN threat status are available in the IUCN Red List database (https://www.iucnredlist.org/). RLS data for species lists and some trait information are available through an online portal accessible through (http://www.reeflifesurvey.com), with additional trait data available on request by using the contact form. For each species, we provide aesthetic values predicted in the present study (S3 Data) and web links to original photographic material (S4 Data). Images free of copyright can be provided on request. Other datasets used in this study (extraction

from the online survey and images features analysis) and all code used for the analysis and figures are available from the GitHub Repository: https://github.com/nmouquet/RLS_AESTHE.

## Supporting information

**S1 File. Supporting information methods and analysis with associated figures and tables.** Text A. Image features analysis. Text B. Image sampling strategy. Text C. Sociocultural background. Text D. Elo scores. Text E. Deep learning algorithm. Text F. Relationship between features and aesthetic values. Text G. Phylogenetic analysis. Text H. Ecological traits. Text I. Mean aesthetic value. Fig A. Representation of the three-dimensional CIELAB space. Fig B. Cluster analysis performed for *Holacanthus ciliaris*. Fig C. HSV color space. Fig D. Illustration of the analysis led with the Momocs package. Fig E. Projection of the 4,881 images of our collection on the 2 first axes of the PCA on the images' features. Fig F. Questionnaire on the sociocultural background of the judges. Fig G. Number of judges per country. The map was obtained from rworldmap v1.3.6 R package, which uses Natural Earth as base layer (https://www.naturalearthdata.com/downloads/10m-physical-vectors/). Fig H. Summary of sociocultural background of the 13,000 judges. Fig I. Variation of the Elo scores of the images as matches accumulate. Fig J. Linear regression between the Elo scores for the 21 images in common between the 2 surveys. Fig K. Effect of the size of the input images on the performances of the model. Fig L. Architecture of ResNet50 modified to predict the aesthetic values. Fig M. Relationship between the Evolutionary Distinctiveness of species (in MY) and their aesthetic value. Fig N. Mean aesthetic values of families with more than 10 species presented in decreasing order. Fig O. Comparison between the aesthetic value of the fish species and their ecological traits. Fig P. Number of images per species. Fig Q. Relationship between the aesthetic values computed using the maximum and the mean values. Fig R. Phylogenetic history and ecological originality with mean aesthetic values. Fig S. Conservation status with mean aesthetic values. Table A. Analysis of deviance in the generalized linear mixed model. Table B. Pagel's λ. Table C. List of the ecological traits used with their nature and modalities. Table D. *P* values of the Tukey's tests.
(DOCX)

**S1 Data. Copyright information for the images used in the figures.**
(XLSX)

**S2 Data. Species included in the study.**
(XLSX)

**S3 Data. Aesthetic values for the 2,417 species of the study.**
(XLSX)

**S4 Data. URL links for the 4,881 original images used in the study.**
(XLSX)

## Acknowledgments

We would like to thank the thousands of volunteers who participated in the online questionnaire, and Graham J. Edgar, Andrew Green, Natali Lazzari, and Ian Shaw for providing the fishes silhouettes used in our figures.

## Author Contributions

**Conceptualization:** Nicolas Mouquet.

**Data curation:** Juliette Langlois, Florian Baletaud, Michel Kulbicki, Aliénor Stahl, Rick D. Stuart Smith, Anne-Sophie Tribot, Nicolas Mouquet.

**Formal analysis:** Juliette Langlois, François Guilhaumon, Florian Baletaud, Cédric De Almeida Braga, Valentine Fleuré, Nicolas Loiseau, Aliénor Stahl, Nicolas Mouquet.

**Funding acquisition:** David Mouillot, Nicolas Mouquet.

**Methodology:** François Guilhaumon, Julien P. Renoult, Anne-Sophie Tribot, Nicolas Mouquet.

**Project administration:** Nicolas Mouquet.

**Visualization:** Juliette Langlois, Nicolas Casajus, Nicolas Mouquet.

**Writing – original draft:** Juliette Langlois, Nicolas Mouquet.

**Writing – review & editing:** Juliette Langlois, François Guilhaumon, Florian Baletaud, Nicolas Casajus, Cédric De Almeida Braga, Valentine Fleuré, Michel Kulbicki, Nicolas Loiseau, David Mouillot, Julien P. Renoult, Aliénor Stahl, Rick D. Stuart Smith, Anne-Sophie Tribot.

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
