## [Editor Report · Decision Letter 0]

2 Nov 2021

Dear Dr Mouquet, 

Thank you for submitting your manuscript entitled "Global mismatch between the aesthetic value of reef fishes and their conservation priorities" for consideration as a Research Article by PLOS Biology.

Your manuscript has now been evaluated by the PLOS Biology editorial staff, as well as by an academic editor with relevant expertise, and I'm writing to let you know that we would like to send your submission out for external peer review.

Once your full submission is complete, your paper will undergo a series of checks in preparation for peer review. Once your manuscript has passed the checks it will be sent out for review. 

If your manuscript has been previously reviewed at another journal, PLOS Biology is willing to work with those reviews in order to avoid re-starting the process. Submission of the previous reviews is entirely optional and our ability to use them effectively will depend on the willingness of the previous journal to confirm the content of the reports and share the reviewer identities. Please note that we reserve the right to invite additional reviewers if we consider that additional/independent reviewers are needed, although we aim to avoid this as far as possible. In our experience, working with previous reviews does save time. 

If you would like to send your previous reviewer reports to us, please specify this in the cover letter, mentioning the name of the previous journal and the manuscript ID the study was given, and include a point-by-point response to reviewers that details how you have or plan to address the reviewers' concerns. Please contact me at the email that can be found below my signature if you have questions. 

Please re-submit your manuscript within two working days, i.e. by Nov 04 2021 11:59PM.

Kind regards,

Roli Roberts

Roland Roberts

Senior Editor

PLOS Biology

rroberts@plos.org

---

## [Decision Letter · Decision Letter 1]

23 Dec 2021

Dear Dr Mouquet,

Thank you for submitting your manuscript "Global mismatch between the aesthetic value of reef fishes and their conservation priorities" for consideration as a Research Article at PLOS Biology. Your manuscript has been evaluated by the PLOS Biology editors, an Academic Editor with relevant expertise, and by four independent reviewers.

You’ll see that all four reviewers are quite positive about the study; however, they all raise a number of concerns, the most serious of which seems to be that raised by both reviewers #1 and #2 about the representativeness of the questionnaire respondents. The remaining issues are largely textual and/or presentational. These must all be addressed for further consideration.

IMPORTANT: The Academic Editor also wants you to address in your Discussion a point that s/he mentioned to me during the initial assessment, namely: "What can we do with that information? How does that address the challenge that a panda will always get more conservation attention than some rare ugly rat?"

In light of the reviews (below), we are pleased to offer you the opportunity to address the [comments/remaining points] from the reviewers in a revised version that we anticipate should not take you very long. We will then assess your revised manuscript and your response to the reviewers' comments and we may consult the reviewers again.

We expect to receive your revised manuscript within 2 months.

**IMPORTANT - SUBMITTING YOUR REVISION**

*Resubmission Checklist*

*Published Peer Review*

*PLOS Data Policy*

*Blot and Gel Data Policy*

Sincerely,

Roli Roberts

Roland Roberts

Senior Editor

PLOS Biology

rroberts@plos.org

REVIEWERS' COMMENTS:

Reviewer #1:

In their paper titled "Global mismatch between the aesthetic value of reef fishes and their conservation priorities" (MS PBIOLOGY-D-21-02764R1), authors have presented an approach for analysis of aesthetic values and its application to ray-finned reef fishes.

I found it an excellent and important study, considering both the implications of their results and the great potential value of the presented approach, with wide applicability in the fields of ecology, conservation, and social sciences. The manuscript is generally well developed, and the study should be relevant and of interest for a wider audience. 

There are however several issues that need to be addressed before I would encourage publication of the study.

Main comments

1. Probably the major problem in the manuscript is the fact that one part of the species were excluded from the analysis, if I understood excluded species were mainly flatfishes and seahorses. As I understood, the reason for their exclusion was that their specific morphology made them behave like outliers in the dataset that complicated the analysis? While this issue has been mostly glossed over, it is one of the main problems in the study, because it may be a source of a bias. It is important to explain and address this issue properly: 1) to provide more detailed arguments for this decision; 2) to provide exact number of species excluded from each species group; 3) to address this issue in Discussion, both in the section with caveats where it is currently not mentioned at all, as well as when discussing results. While being closer with some traits to species classified as having low aesthetic value, seahorses are considered highly attractive. I would therefore expect that they would rank highly with their aesthetic value, and they are at the same time endangered, which is an important caveat with respect to the conclusions drawn.

2. Study included only ray-finned fishes, which is important to acknowledge better, including in the title and abstract (i.e. instead of "reef fishes" to use "ray-finned reef fishes"). For example, reef sharks are considered as highly charismatic and likely to get high aesthetic scores, while their morphology (body shape and coloration) match traits identified here as non-aesthetic.

3. Aesthetic preferences and perceptions can considerably vary regionally, culturally, among different social and demographic groups, as well as temporally. This needs to be better addressed in the manuscript, including relevant literature, especially considering that the pool of respondents is strongly biased towards France, as well as Europe and the Global North.

4. There is insufficient information about how the questionnaire was distributed, whether there were financial or other incentives to distribute it, and especially the languages in which it was provided, to understand the reasons for such strongly biased pool of respondents. 

5. I suspect based on information in Lines 491-492 that the questionnaire dissemination was improperly planned, aiming mainly to respondents in France, and using academic or similar mailing lists, which resulted in such skewed pool of respondents, both spatially and with respect to education level, mostly failing to reach other countries, continents and regions, as well as general population, beyond the highly educated respondents. This is one of the major weaknesses of the study, and needs to be better acknowledged and discussed in the paper.

6. Authors have pooled species that are categorized by the IUCN Red List as Not Evaluated (NE) and Data Deficient (DD) into a single group for analyses, "Not Evaluated". However, there is a considerable literature showing that Data Deficient species tend to be overall highly threatened (i.e., their traits, low population size, small ranges, and other characteristics that are making them highly threatened are also making them poorly studied and consequently data deficient). For this reason, DD species do not really belong together with NE, especially considering that comparisons of NE and TH groups were used to conclude that threatened species have lower aesthetic value. I do not think or suggest that it is necessary to redo complete analysis with DD species considered separately, but authors should at least acknowledge this issue and refer to the literature that addressed the issue of the threat status of data deficient species.

7. It would be good to include a table, at least in the supplementary material, which would present number of species and photos from different sources, used in each step (e.g. initial number of species, species excluded, etc.), to make it easier to understand the dataset used.

Other comments

8. Referring to respondents of the questionnaire as "judges" makes those sections confusing. It would be clearer and more appropriate to refer to them as "respondents".

9. There is some repetition in Results section of the information that is already presented in Methods section. Most of the subchapters in Results begin with description of methods. Such sections should be deleted, or at least condensed to reduce repetitiveness of the text.

10. Line 132, "Figure 1A" - "A" shouldn't be capitalized

11. It is unnecessary to refer to names of variables that were used during the analysis (e.g. "CL_cie_d_mean"), these names are not relevant and make it more difficult to follow the text (e.g. in Lines 225-228 and in Figure 2). Instead of those names it would be better to simply refer to the actual variable (e.g. "color heterogeneity")

12. Figure 3 caption (Line 269) mentions "ecological distinctiveness", while the X axis in the figure is named "Functional distinctiveness". Terms should be consistently used in the study.

13. Some species names in Figure 4 caption appear twice (Pterois miles and Cephalopholis sexmaculata in Lines 283-284). Just in case, please check this also in other parts of the manuscript, figures and supplementary material, as well as spelling of all species names, as it is one of common source of errors.

14. Line 324 - "Least Concerned" should be corrected to "Least Concern"

15. Line 400 - "though" should be corrected to "through"

16. Line 545 - it is left unclear why were these 137 species from FishBase not already covered by the original dataset of 2280 species, why were they missing from RLS?

17. Line 593, "Fig S5" - Which figure is this? Figures in the supplementary material are all named with capital letters.

18. Line 609 - "images" should be in singular

19. Line 763 - Sentence should be corrected to "Data on IUCN threat status are available in the IUCN Red List database".

20. S1 Fig G - legend of the inset figure needs more values on the color scale, not just the maximum and minimum. It should also not be log transformed, as it does not present clearly bias in spatial coverage.

21. S1 Fig. H - Y axis categories need to be made clear (e.g. education, place, distance, etc.)

Reviewer #2:

Langlois et al. present a compelling and important study connecting aesthetic and ecological values across almost 2500 coral reef fish species. The researchers conduct a large-scale online image analysis survey ('competing' paired images), as well as previous data, to score >300 species and then use these images/scores to train an AI algorithm, allowing rapid predictions of aesthetic score for the remaining ~2000 species. The paper first confirms that certain traits such as color intensity and diversity and disc-like shapes are favored by observers. It then goes on to provide novel evidence that aesthetic score declines with species age, as well as evolutionary and functional distinctiveness. The paper further provides a complementary ancestral state analysis to show the evolution and pattern of aesthetic value across coral reef fish in the Tree of Life. Finally, the team show aesthetic score is slightly lower in threatened and commercially important species. As conservation, public/political, and research attention is greatly affected by aesthetic qualities of biota, this research is highly important and stands to make a large impact. 

I have a concern about the representativeness of the population of people surveyed. The paper addresses the aesthetic value of global coral reef fishes, and does so by distributing online surveys through existing networks. Does this create a cultural bias in the sampled population with respect to the global population, perhaps especially neglecting those people interacting with coral reef fish through subsistence fishing or other livelihoods? Furthermore, it is indicated that the selection of fish species ensured that they were those encountered by divers, but the diving theme is not further developed or linked to the sampled population. Generally I would like to see more consideration and discussion of the rationale and global representativeness of the sampled population, and a more explicit explanation of whether the focus is on fish encountered by SCUBA divers (as hinted at), and therefore perhaps a population of potential tourist SCUBA divers, or whether the study and results are meant to be more broadly applicable across populations around the world and with respect to different ways in which people worldwide interact with coral reef fish, including in subsistence settings where some of the highest rated fishes are fished. 

Generally the AI algorithm does well at predicting aesthetic scores (Fig. 1B). However, in perhaps the top 5 or 10% of the range of evaluated aesthetic scores the algorithm does much more poorly. After X = 1750 all of the points are above the line, meaning AI was over-estimating the aesthetic values. I don't expect this to influence the results (looking at the other figures) but checking these points don't change the later models might be useful. 

The statistical models relating aesthetic value with species age, ED and functional distinctiveness do not appear to account for phylogenetic relatedness, even though aesthetic value shows a strong phylogenetic signal. I realize that it is not always relevant to account for phylogeny in functional analyses, but it would be useful if the authors could comment on the rationale for leaving out phylogeny from the functional analysis, and what the implications are regarding inference? Does this limit our approach to conclude whether the differences are being driven by a couple of key contrasting families (e.g., beautiful butterflyfishes vs. brutish barracudas?), as opposed to a more general effect across the Tree? 

Red squiggly spell-check lines in 1A should be removed

Reviewer #3:

[identifies himself as Uri Roll]

I found this contribution to be interesting, novel, and important. I have but few minor comments to it. 

I thought that the abstract could benefit from specifically mentioning that you used fishes' images (perhaps also mention from readily available online sources)

Perhaps mention in the methods the socio-cultural variables you explored even broadly (and not just in the supp.)

In figure 2 try to give the actual names of features and not their abbreviations.

Generally, try to reduce a bit the repetition between the methods and results section.

Reviewer #4: 

[see attached file for fully formatted version]

Thank you for this opportunity to review this manuscript. The review below is also attached as a PDF file through the reviewing platform.

Review PBIOLOGY-D-21-02764_R1

Global mismatch between the aesthetic value of reef fishes and their conservation priorities

Thank you for this opportunity to review this manuscript entitled "Global mismatch between the aesthetic value of reef fishes and their conservation priorities" for Plos Biology. This manuscript presents an elegant way to evaluate the aesthetic value of reef fishes and compares this value to several attributes of the fish species, like their conservation status, their ecological traits and ecological distinctiveness. It relies on unparalleled datasets on reef fish images on one side and corresponding evaluation from the broad public (>13,000 respondents) on the other side. It thus makes for an impressive social-ecological assessment of >4,000 species of fish underpinning complementary regulating, material, and most importantly (and innovatively in the present study) non-material Nature's Contributions to People. The paper is very well written and easy to read, and the figures nicely convey the main messages of the paper. I would like to simply congratulate the whole team of authors for their very innovative social-ecological work dedicated to find ways to quantify how human societies value their environment. I have a list of comments and suggestions that pertain to the different sections of the manuscript, that I hope will help the team improve some arguments in the introduction and discussion, and clarify a few aspects of the methods and results.

COMMENT ON THE TITLE

The title advertises an evaluation of the "global mismatches" between aesthetics and conservation threat. In the valuation literature, the term mismatch is widely used, in particular to describe the mismatches between the supply and demand for an object of valuation. In this manuscript, the term is not used in the text, neither in the introduction nor in the discussion, and having it in the title may be misleading for the readership. Besides, you also use the term "match" to describe a pair of images being compared to one another, and the term "mismatch" may also be confusing. I would suggest to slightly change the title to avoid this potential double confusion. In addition, the mismatch between the aesthetic value and the conservation status is postulated to be a concern, but this may not be the case (see my comments on the discussion). 

COMMENTS ON THE INTRODUCTION

L68: what do you mean by "means of NCP" ? consider rephrasing?

L95-96: can you be a bit more specific here and explain what you mean by "visual feature"?

L.101: the sentence "to accurately identify patterns and predict values on images" is confusing. What "value" is being predicted and to what is this value assigned? As this sentence is key to describing the advantages of CNN, I would recommend to be crystal clear on what CNN does that other approaches don't, and how it operates in more concrete terms.

L.108: "public members" feels odd (but note that I'm not a native speaker). I would suggest "13,000 respondents from the broad public".

L.112: typo "evaluated"

L.112-119: In this paragraph, you justify the use of coral reef fishes as a study object by demonstrating their importance first for material NCP (economic activity and industry), regulating NCP (functioning of coral reef), and then only for non-material NCP (aesthetic values). I believe it would make more sense to reverse the argument, as you nicely explain in the paragraphs above that assessments of aesthetic values and, more broadly, non-material NCP are lagging behind material and regulating ones. I would therefore recommend to start with their importance in non-material terms with greater emphasis (i.e. what makes them the perfect object of study to investigate non-material NCP and aesthetic value in particular; diversity of morphologies, aesthetic values, opportunities for recreation, and probably more?), then the regulating (their contribution to the functioning of ecosystem) and then their contribution to material NCP. Besides, there is a slight contradiction between the introduction and the interpretation of the results, because reef fish species are presented as key components for economic activity (L.113-115) to justify using them as a study object, when "only" a fifth of them according to Fig.5b are used for the fishing industry. Rearranging the paragraph would restore the balance in the argumentation a bit.

COMMENTS ON THE RESULTS

Fi.1a panel (ii): the screen capture of the box capture the red underlining of Word/Powerpoint, which should be removed.

L.131-142 & L167-169: these sentences rather belong to material and methods, but I let the authors and/or the editor appreciate whether they are essential to the presentation of the results.

L.138: the term "judge" is rather unusual. I would recommend using "respondent" and stick to it throughout the text and in the supplement

L.193: specific "predicted aesthetic values"; the values have not been described so far in the text, and I believe this is an important aspect to clarify. The valuation literature is full of examples of different valuation methods, and giving a few words on the Elo method would be critical, in my humble opinion.

Fig.2a: I would suggest to have clearer labels for each of the 9 image features in the figure. The labels are rather cryptic in their current form and do not help the reading.

L212-219: why don't you provide formal testing of the relationship between aesthetic value and both PCA axes, instead of only a visual interpretation of the patterns in Fig. 2b?

Fig.4: the color gradient should be displayed as a legend on the figure itself.

COMMENTS ON THE DISCUSSION

The paper highlights that the aesthetic value of fish is inversely proportional to their threat status and to their ecological originality/distinctiveness/usefulness (e.g.L.293). Playing the advocates' devil, isn't that a good thing that the aesthetic, intangibles values associated non-material NCP are complementary to the material and regulating NCP provided by those fish? This complementarity is an exciting finding in my opinion, as a broad spectrum of fish species are valued for complementary reasons, which makes the case for paying attention to multiple aspects to preserve them: ecological, aesthetic, and economic. To frame my question differently: what would happen if the fish species that were appreciated for their aesthetics were also highly valued by the industry, and were the most original ones and essential to the functioning of ecosystems? Worse, if they were also threatened, there would be a greater risk of an anthropogenic allee effect, whereby rare + beautiful species would suffer from an even greater demand from the broad public.

L.443-446: how is this statement backed-up by the results, and what do you mean by "less well-represented in the assessments"? I would recommend greater clarity in this statement, as it is not straightforward that data deficient species are less attractive than species present in the IUCN assessments. Besides, isn't this contradictory to the earlier statements in this paragraph about charismatic and aesthetically attractive mammal species receiving greater conservation attention (L.441-442)? Here, you demonstrate that the most aesthetically pleasing species are not particularly threatened, and will therefore not necessarily need greater conservation attention.

COMMENTS ON THE METHODS

L. 566-567: given the fact that the evaluation of the 9 features in an image is based on saturation and lightness (L.665), how did you make sure that your photoshop correction did not bias the results in any way? How many pictures did you correct that way? I do acknowledge that some corrections may have been necessary, but given the later evaluation of the image characteristics, why not discarding the images that were deemed in need of a photoshop correction?

I could not find the information on the number of "matches" undertaken by each species in the dataset. I don't doubt that with over 13,000 respondents evaluating 30 matches, the distribution of matches over all species is hypothetically even, but I was curious to know if this was the case. 

It is a bit surprising that no ethical statement is provided concerning the use of a social survey. Usually, an informed consent is provided by the respondent, acknowledging that the participant undertakes the survey on a voluntary basis, that anonymity is safeguarded (this is the case here), that the respondent can opt out at no cost, and that the objective of the research has been clearly stated. Was such a consent from the participants clearly formulated?

---

## [Editor Report · Decision Letter 2]

6 Apr 2022

Dear Dr Mouquet,

Thank you for submitting your revised Research Article entitled "Global mismatch between the aesthetic value of reef fishes and their conservation priorities" for publication in PLOS Biology. The Academic Editor and I have now assessed your responses and revisions.

Based on this assessment, we will probably accept this manuscript for publication, provided you satisfactorily address the remaining points raised by the Academic Editor, and the following data and other policy-related requests.

IMPORTANT: Please attend to the following:

a) Please re-arrange your title to incorporate an active verb. We suggest the following: "The aesthetic value of reef fishes is globally mismatched to their conservation priorities."

b) Please address the minor requests from the Academic Editor, pasted into the bottom of this email.

c) The Academic Editor thinks (and we agree) that it would be extremely useful for future readers to make the peer-review history of your manuscript accessible, so they can read and assess your thoughtful answers to the reviewers. You will be offered this option later on during the production process, and we ask that you take it.

d) When I was reading about your questionnaire in the Methods section, I wondered how the participants were advised regarding the nature of the project and their consent. In the Supplementary Information I found the following paragraph: "Note that the internet questionnaire we used contained an introduction that clearly stated the objective of the research and an ethical statement to guarantee anonymity to the respondents; we provided each respondent with an anonymity identification number which could be used to opt out at no cost following the French Data Protection Authority (CNIL) recommendation." Please include this helpful information in the main paper.

e) Please address my Data Policy requests below; specifically, we need you to supply the numerical values underlying Figs 1B, 2AB, 3, 4, 5, S1E, S1G, S1H, S1I, S1J, S1M, S1N, S1O, S1P, S1Q, S1R, S1S, S1T. Please also cite the location of the data clearly in each relevant main and supplementary Fig legend, e.g. if, for example, the Figs can all be generated using the data and code in your Gthub deposition, you should write “Data and code required to generate this Figure can be found in https://github.com/nmouquet/RLS_AESTHE”.

We expect to receive your revised manuscript within two weeks. 

*Published Peer Review History*

*Press*

Sincerely,

Roli Roberts

Senior Editor,

rroberts@plos.org,

PLOS Biology

DATA POLICY:

Regardless of the method selected, please ensure that you provide the individual numerical values that underlie the summary data displayed in the following figure panels as they are essential for readers to assess your analysis and to reproduce it: Figs 1B, 2AB, 3, 4, 5, S1E, S1G, S1H, S1I, S1J, S1M, S1N, S1O, S1P, S1Q, S1R, S1S, S1T. NOTE: the numerical data provided should include all replicates AND the way in which the plotted mean and errors were derived (it should not present only the mean/average values).

DATA NOT SHOWN?

COMMENTS FROM THE ACADEMIC EDITOR:

I caught a few very minor points on my reread:

* "judges" still appears in Fig. 1 despite the authors saying that they have changed all instances to "respondents” in response to one of the reviewer’s comments

* Line 229 of the SI: It would be good to clarify explicitly that the +/- refers to a standard deviation on first usage (I presume it does?). The same issue appears in the Main Text, e.g. line 156, 192, 248-250 (where by the way there is no space after the +/- but there is on line 252).

* There is an extra full stop on line 104 after "variables" that should be removed.

* I agree with R4 on line 143, which I don’t think the authors have addressed satisfactorily. I think that instead of "through Elo scores computation" the authors should say something like "by computing a metric that considers ... few words go here to explain what a Elo score is to a general reader".

---

## [Editor Report · Decision Letter 3]

21 Apr 2022

Dear Dr Mouquet,

On behalf of my colleagues and the Academic Editor, Andrew Tanentzap, I'm pleased to say that we can in principle accept your Research Article "The aesthetic value of reef fishes is globally mismatched to their conservation priorities" for publication in PLOS Biology, provided you address any remaining formatting and reporting issues. These will be detailed in an email that will follow this letter and that you will usually receive within 2-3 business days, during which time no action is required from you. Please note that we will not be able to formally accept your manuscript and schedule it for publication until you have completed any requested changes.

As mentioned previously, we think that it would be extremely useful for future readers to make the peer-review history of your manuscript accessible, so they can read and assess your thoughtful answers to the reviewers. You will be offered this option later on during the production process, and we ask that you take it (we note that you are happy to do this).

Sincerely, 

Roli Roberts

Roland G Roberts, PhD 

Senior Editor 

PLOS Biology

rroberts@plos.org